# Chiral Indolizinium Salts Derived from 2-Pyridinecarbaldehyde—First Diastereoselective Syntheses of (−)-1-*epi-*lentiginosine

**DOI:** 10.3390/molecules28093719

**Published:** 2023-04-25

**Authors:** Hisami Rodriguez-Matsui, David M. Aparicio, María L. Orea, Jorge R. Juárez, Victor Gómez-Calvario, Dino Gnecco, Alan Carrasco-Carballo, Joel L. Terán

**Affiliations:** 1Centro de Química, Instituto de Ciencias, Benemérita Universidad Autónoma de Puebla, Edif. IC9 Complejo de Ciencias, C.U., Puebla 72570, Mexico; hisami_matsui@hotmail.com (H.R.-M.); davidmiguel.aparicio@correo.buap.mx (D.M.A.); maria.orea@correo.buap.mx (M.L.O.); jorge.juarez@correo.buap.mx (J.R.J.); victor.gomez@correo.buap.mx (V.G.-C.); dino.gnecco@correo.buap.mx (D.G.); 2Laboratorio de Elucidación y Síntesis en Química Orgánica, Benemérita Universidad Autónoma de Puebla, C.U., Puebla 72570, Mexico; alan.carrascoc@correo.buap.mx

**Keywords:** diastereoselective synthesis, (−)-1-*epi*-lentiginosine, *trans*-epoxyamides, indolizinium salts

## Abstract

The first diastereoselective synthesis of (−)-1-*epi*-lentiginosine from a common chiral *trans*-epoxyamide derived from 2-pyridincarbaldehyde is reported. This methodology involves a sequential oxirane ring opening and intramolecular 5-*exo*-*tet* cyclization of tosylate *trans*-epoxyalcohol to afford a diastereomeric mixture of indolizinium salts in a one-pot fashion, followed by regio- and diastereospecific pyridinium ring reduction.

## 1. Introduction

Hydroxylated indolizidines, such as (+)-castanospermine, (‒)-swainsonine, (+)-lentiginosine and (‒)-*epi*-lentiginosine, are widely found in plants and microorganisms, and they also constitute a class of azasugars that exhibit potent and selective glycosidase inhibitory activities [1,2]. Specifically, lentiginosine is known not only to be a significant inhibitor of amyloglycosidases but also to have excellent anti-HIV, anti-tumor and immunomodulatory activities (Figure 1) [1,3,4,5,6]. Therefore, several syntheses using a chiral pool [7,8,9,10,11,12,13,14,15,16,17,18,19,20,21,22,23,24,25,26,27] or enantio- or diastereoselective [28,29,30,31,32,33,34] approaches to lentiginosine have been reported.

Although there are few reports of the use of pyridine derivatives in the synthesis of lentiginosine, this heterocycle has proved to be a valuable starting material for the synthesis of this indolizidine. For example, Zhou reported the use of ethyl 3-(pyridine-2-yl)acrylate *N*-oxide obtained from picolinaldehyde via the Wittig reaction. Subsequently, the asymmetric dihydroxylation of heteroaromatic acrylate affords the key intermediate for the synthesis of (+)-lentiginosine at an overall yield of 16% [31]. Fruit et al. reported a synthesis of (−)-lentiginosine and its epimers starting from 2-bromopyridine condensed with an enantiomerically pure (*R*)-2,3-*O*-isopropylidene glyceraldehyde prepared from D-mannitol. The key step involved quaternization of a completely unprotected pyridinium-polyol unit using the Mitsunobu methodology, followed by PtO_2_-catalyzed diastereoselective hydrogenation of the pyridinium ring to give the desired dihydroxyindolizidine [35]. Finally, Brandi et al. reported the synthesis of (±)-lentiginosine starting from 1-(2-Pyridyl)-2-propen-1-ol. In this methodology, the main steps were a domino process involving the electrophilic addition of bromine to the propen-1-ol derivative and the cyclization of the bromonium ion to give the corresponding indolizinium salt followed by a diastereoselective reduction, resulting in a diastereomeric mixture of tetrahydro derivatives. Nucleophilic substitution (via elimination and addition) finally yields (±)-lentiginosine [36].

On the other hand, we showed, in a previous report, that chiral *N*-phenylglycinol-derived 2,4-disubstituted oxazolidines are excellent chiral auxiliaries for the asymmetric condensation of amide-stabilized sulfonium ylides with aldehydes to access at *trans*-glycidic amides in high diastereoselectivity [37]. Following this work, and in order to highlight our diastereoselective epoxidation methodology, we develop the total synthesis of lentiginosine using *trans*-glycidic amide as starting material derived from chiral oxazolidine sulfur ylide and 2-pyridincarbaldehyde (Figure 1).

## 2. Results

Our retrosynthetic analysis is outlined in Figure 2. We envisioned that lentiginosine could be accessed via a diastereoselective reduction of indolizinium salt I which can be generated by intramolecular cyclization reaction of diol II. The intermediate II could be obtained via a reductive removal of the chiral auxiliary of the glycidic amide III and the opening of the oxirane function. III can be prepared from commercially available 2-pyridine carbaldehyde and the chiral oxazolidine sulfur ylide derived from (*R*)-(−)-2-phenylglycinol (Figure 2).

First, we prepared the chiral oxazolidine sulfur ylide **1** and its subsequent condensation with 2-picolinaldehyde to give the desired glycidic amide **2** following our previously reported reaction strategy [37]. Compound **2(a+b)** was obtained in a high chemical and stereochemical yield (85% yield, 95:5 *d.r*.). The major diastereoisomer **2a** was separated and we assumed the configuration of the new stereogenic centers as (2*R*, 3*S*) according to our previous report (Figure 3).

With the major diastereoisomer, **2a** in hand and continuing with our retrosynthetic plan, the next step consisted of the reductive removal of the chiral auxiliary. Unfortunately, all attempts to obtain the desired epoxyalcohol were unsuccessful (LiBH_4_, super-hydride^®^, Red-Al^®^ were tested), and the oxirane intermediate **2a** was completely degraded.

In this way, and as mentioned earlier, Brandi et al. [36] reported a racemic synthesis of lentiginosine via cyclization of the bromohydrin intermediate derived from 1-(pyridin-2-yl)prop-2-en-1-ol, for which epoxyamide **2a** was treated with HBr to access the corresponding halohydrin. Pleasingly, the corresponding bromohydrin **3** was obtained when intermediate **2a** was dissolved in DCM, and HBr (three drops of aqueous 45% solution) was added at room temperature. This product was detected by NMR spectroscopy as a mixture of bromohydrin rotamers since signals coalesced at a higher temperature (two doublets around 4.99 and 4.73 ppm with *J* = 5.5 Hz) (see Appendix A).

Crystallized compound **3** enabled the unambiguous determination of the absolute stereochemistry of the new stereogenic centers as (2*S*,3*S*) and confirmed that a regio- and diastereospecific oxirane opening reaction occurs (Figure 2) [38]. To improve the reaction yield, screening experiments were performed. The best result was obtained when epoxyamide **2a** was treated with HBr in CHCl_3_ at room temperature (entry 2, Table 1). Compound **3** was obtained in 98% yield after chromatographic column purification.

Then, reductive removal of the chiral auxiliary was performed. Bromohydrin **3** was reacted with LiBH_4_ under ultrasonic activation conditions. Not only the chiral auxiliary was removed but 73% of the corresponding epoxyalcohol **4** was also obtained, via an intramolecular nucleophilic substitution, as a specific diastereoisomer (Figure 4).

Tosylation of epoxy alcohol **4** afforded the desired compound **5** in 94% yield after purification by column chromatography. Immediately, compound **5** was treated with hydrogen halide to promote the formation of the halohydrin. All attempts resulted in the formation of the corresponding diastereomeric mixture of indolizinium salts **6(a+b)**, obtained by a domino process involving a regiospecific oxirane opening and an intramolecular nucleophilic cyclization reaction favoring the formation of the *cis* isomer (*J_H_*_1,*H*2_ = 4.7 Hz) as a result of the more favorable nucleophilic attack on the oxirane ring on the backside. The use of HBr gave the corresponding indolizinium salts **6(a+b)** in 80:20 *dr*. The best diastereoselectivity was obtained with the use of HCl, which gave the desired diastereomeric mixture of indolizinium salts **7(a+b)** in *dr* = 90:10 (the diastereomeric ratio was measured directly from the NMR spectra of the crude reaction). Finally, the use of HI resulted in decreased diastereoselectivity (Figure 5).

In order to extend the scope of this domino reaction and to take into account that this reaction occurs in an acidic medium, we decided to investigate this reaction with other nucleophiles catalyzed with Lewis acid. After testing with various Lewis acids (Cu(OTf)_2_, BF_3_•OEt_2_, etc.), this reaction occurred in the presence of Ytterbium(III)triflate (10 mol%) [39] with a nucleophile (benzylic alcohol or H_2_O) in high chemical and stereochemical yields (the diastereomeric ratio was measured directly from the NMR spectra of the crude reaction) (Figure 6).

Unfortunately, all attempts to separate the diastereomeric indolizinium salts were unsuccessful. Therefore, we turned our attention to the pyridinium ring reduction without purification, first using the corresponding inseparable diastereomeric chlorohydrin salt **7(a+b)**.

Catalytic hydrogenation of **7(a+b)** in the presence of Pd/C afforded a complex mixture and recovery of the starting material despite longer reaction times and different catalysts being explored. Pleasingly, the use of NaBH_4_ leads to the tetrahydro derivatives **11(a+b)**. This mixture was separated by column chromatography, and each diastereoisomer was crystallized, which allowed for the determination of its absolute stereochemistry via X-ray crystallography [38]. Based on the results obtained, we propose that the hydride addition occurs in a diastereospecific manner from the less hindered face of the bicyclic indolizinium salt (Figure 7).

Using the major diastereoisomer **11a**, we attempted to complete the total synthesis of (+)-lentiginosine. Although the catalytic hydrogenation cleanly affords indolizidine **12a** [38], the absolute configuration of which was determined via X-ray diffraction analysis, the substitution of a chlorine atom by OH was unsuccessful (Figure 8).

Therefore, the total synthesis of (−)-*epi*-1-lentiginosine was carried out starting from a mixture of indolizinium salts **10(a+b),** which was subjected to acidic hydrogenation conditions to afford the desired (−)-1-*epi*-lentiginosine in 87% yield after chromatographic purification (Figure 9).

## 3. Materials and Methods

### 3.1. General Information

All reagents and solvents were purchased from commercial sources. The ^1^H and ^13^C spectra were determined with a Bruker Avance III Spectrometer (CDCl_3_ or CD_3_OD solvents) operating at 500 and 125 MHz, respectively. The chemical shifts were reported in parts per million (ppm), downfield from SiMe4 (δ 0.0) and relative to the signal of chloroform-d (δ 7.26, singlet). Multiplicities were afforded as: s (singlet); d (doublet); t (triplet); q (quartet); dd (doublets of doublet); ddd (doublet of doublets of doublets); or m (multiplets). The number of protons for a given resonance is indicated by nH. Coupling constants were reported as a *J* value in Hz. Thin layer chromatography (TLC) was used to monitor the reaction on Merck 60 F254 precoated silica gel plate (0.2 mm thickness). Optical rotations were determined at room temperature with a Perkin-Elmer 341 polarimeter, using a 1 dm cell with a total volume of 1 mL, and are referenced to the D-line of sodium. Mass spectra were recorded with a JEOL Station JMS-700 instrument at a voltage of 70 eV. X-ray diffraction analysis was performed on a diffractometer STOE Stadivari using Ag-Kα radiation (λ = 0.56083 Å, AXO micro-source) and equipped with a Dectris Pilatus-100 K detector. Intensities were collected at 295 K, and structures were refined using the current release of SHELXL (2018/3). The products were purified by column chromatography on silica gel 60 (63–200 nm).

### 3.2. General Procedures

*Synthesis of (2S,3S)-2-bromo-3-hydroxy-1-((2R,4R)-4-phenyl-2-propyloxazolidin-3-yl)-3-(pyridin-2-yl)propan-1-one, * **3**.

*trans*-epoxyamide **2a** (0.1 g, 1.0 equiv, 0.29 mmol) was dissolved in 1 mL of chloroform at 25 °C; then, 3 drops of HBr (48%) were added, and the reaction mixture was stirred for 40 min. Then, NaHCO_3_ was added, and the resulting reaction mixture was filtered. Finally, the solvents were evaporated. Product **3** was crystallized in a mixture of petroleum ether: DCM (70:30). Bromohydrin **3** (CCDC: 2181510) was obtained in 98% yield (all spectroscopic details are described in ESI).

*Synthesis of ((2S,3S)-3-(pyridin-2-yl)oxiran-2-yl)methanol,* **4**.

To a solution of bromohydrin **3** (110 mg, 0.26 mmol) in anhydrous THF (0.08 M) under ultrasonic activation at 5 °C, a solution of LiBH_4_ (2 M, THF, 5 equiv.) was added in portions. After 2 h, 1 mL of H_2_O_2_ (30%) was added slowly to the reaction mixture followed by addition of NaOH (3 N) solution. The mixture was stirred overnight at room temperature. The crude reaction was then filtered, and then the organic layer was dried over anhydrous Na_2_SO_4_. After evaporation of the solvent from the filtrate, the residue was subjected to purification by flash column chromatography (SiO_2_, CH_2_Cl_2_/MeOH). Epoxyalcohol **4** was obtained in 73% yield (all spectroscopic details are described in ESI).

*Synthesis of ((2S,3S)-3-(pyridin-2-yl)oxiran-2-yl)methyl 4-methylbenzenesulfonate,* **5**.

Compound **4** was dissolved in DCM, and the resulting solution was cooled to 0 °C; then, Et_3_N (0.405 mmol, 0.058 mL) and DMAP (0.0026 g, 0.021 mmol) were added. The resulting mixture was stirred for 10 min; then, *p*-TsCl (0.062 g, 0.325 mmol) was added in portions and the mixture was stirred for 1 h. Finally, the reaction was quenched by adding a brine solution, and the organic phase was separated and dried over anhydrous Na_2_SO_4_, filtered and the solvent was evaporated yielding the desired tosylated epoxyalcohol **5,** which was obtained in 94% yield after purification by chromatography (silica gel, AcOEt/petroleum ether) (spectroscopic details are described in ESI).


*General procedure for a one-pot regiospecific oxirane opening and intramolecular nucleophilic cyclization reaction*



*With hydrohalic acids*


Compound **5** (0.053 g, 0.173 mmol) was dissolved in CHCl_3_ (1 mL) at room temperature, and then the corresponding hydrohalic acid was injected through a needle into the solution and stirring for 3 h. Finally, the reaction was quenched by adding NaHCO_3_ until pH = 7. Then, the mixture was filtered, and the solvent was evaporated to obtain the corresponding inseparable diastereomeric mixture of indolizinium salt, which was used without purification for the next reaction: [CCDC: 2181511 (**11a**), 2181512 (**11b**)] (spectroscopic details are described in ESI).


*With H_2_O or BnOH as nucleophile*


To a solution of compound **5** (0.079 g, 0.26 mmol) and the corresponding nucleophile (0.029 mL, 0.228 mmol) in 1,4-dioxane (1 mL), ytterbium trifluoromethanesulfonate (0.016 g, 0.1 equiv.) was added under inert atmosphere. The suspension was stirred for 24 h. Finally, the solvent was evaporated and the desired indolizinium salt was precipitated in AcOEt/petroleum ether given the corresponding inseparable diastereomeric mixture of indolizinium salt.


*Diastereospecific reduction of indolizinium salts*


To a stirred solution of the corresponding indolizinium salt (0.084 mmol, 29 mg) in methanol (3 mL) at 0 °C, NaBH_4_ (10 mg, 0.26 mmol) was added slowly. After, the mixture was stirred for 10 min and then a saturated aqueous solution of NaSO_4_ was added. Next, the resulting mixture was filtered through a celite pad. The solution was evaporated, and the residue was purified by flash chromatography to afford the desired hexahydroindolizin-2-ol (spectroscopic details are described in ESI).

*Synthesis of* (−)-1-*epi*-lentiginosine.

To a methanolic solution of the diastereomeric indolizidinium salt **10(a+b**), concentrated HCl (1 drop) was added, and the resulting mixture was hydrogenated at room temperature in the presence of 10% PtO_2_. The reaction was stirred overnight. Then, NaOH (3 M) solution was added, and the resulting crude reaction was extracted with dichloromethane, dried over anhydrous Na_2_SO_4_, filtered, concentrated and finally subjected to purification by flash column chromatography (SiO_2_, CH_2_Cl_2_/MeOH). The desired (−)-1-*epi*-lentiginosine was obtained with 87% yield (spectroscopic details are described in ESI).

## 4. Conclusions

In conclusion, a novel protocol for the diastereoselective synthesis of substituted indolizidinium salts from a common chiral *trans*-epoxiamide is reported. In addition, the first diastereoselective synthesis of (−)-1-*epi*-lentiginosine in only five steps, and a 49.7% overall yield starting from *trans*-epoxyamide **2a** is reported. This new, versatile and diastereoselective access to chiral indolizidine compounds opens the route to the pharmacological investigation of these promising bicyclic cores as well as the design of analogs.

## Data Availability

No applicable.

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
