# Peer review of "Chiral Indolizinium Salts Derived from 2-Pyridinecarbaldehyde—First Diastereoselective Syntheses of (−)-1-epi-lentiginosine"

_molecules, 2023, doi:10.3390/molecules28093719_

Round 1

Reviewer 1 Report

J. L. Terán et al. presented a directed synthesis of (+)-lentiginosine. On the whole, the study carried out deserves the attention of chemists in the field of asymmetric synthesis. This manuscript can be recommended after some corrections and additions.

It is necessary to correct "minor" in Scheme 3. Comment on the region-specificity of the formation of bromohydrins from epoxides. It would be good to explain the conditions (lithium borohydride) in the formation of epoxide 4. How does lithium borohydride work and why does it not reduce the oxirane ring, or the bromine in the starting compound? The CCDC of all compounds should be duplicated in the experimental part of the article file.

Author Response

Reviewer 1.

Comments and Suggestions for Authors

  1. L. Terán et al. presented a directed synthesis of (+)-lentiginosine. On the whole, the study carried out deserves the attention of chemists in the field of asymmetric synthesis. This manuscript can be recommended after some corrections and additions.

Question: It is necessary to correct "minor" in Scheme 3. Comment on the region-specificity of the formation of bromohydrins from epoxides.

Answer: In Scheme 3, we described the diastereoselective synthesis of starting materials and not described the synthesis of bromohydrins.

The following sentence was added in lines 97-98: “(Two doublets around 4.99 and 4.73 ppm with J = 5.5Hz).”

Question: It would be good to explain the conditions (lithium borohydride) in the formation of epoxide 4. How does lithium borohydride work and why does it not reduce the oxirane ring, or the bromine in the starting compound?

Answer: The paragraph was modified by: “Then, reductive remotion of the chiral auxiliary was performed. Bromohydrin 3 was reacted with LiBH4 under ultrasonic activation conditions. Not only the chiral auxiliary was removed, but 73% of the corresponding epoxyalcohol 4, via an intramolecular nucleophilic substitution, was obtained as a specific diastereoisomer (Scheme 4).”

Question: The CCDC of all compounds should be duplicated in the experimental part of the article file.

Answer:  All CCDC numbers were added in the experimental part of the article file.

Reviewer 2 Report

1. Overall this study has some valuable improvement for the synthesis of the chiral natural product. 

2. In supporting Information, if the redundant S28-S38 are substituted by tables of selected bond lengths and angles for the crystal structures, their stereochemistry would be easier understood.

3. In the Materials and Methods section, it is easier to follow if putting name and number of the compound to each paragraph, instead of only listing the type of reactions (such as only  "Regiospecific oxirane opening" title).

Overall the language is acceptable, with some typos, words and grammars needed to be corrected.

for examples:

Line 95, HBr was added: what form of HBr, gaseous or aqueous? 

Line 83, 0 vs 0 o

line 88, 111,116 etc, remotion vs removal.

Line 96, rotaters the since ?

line 98, compound 3 crystallized vs crystalline compound 3, or crystallized compound 3.

Line 113, 4 was obtained vs 4 was also obtained.

line 149, employ vs use; tested vs explored; line 160, despite the vs although; 

S9 typo, temeratura vs temperature.

Author Response

Reviewer 2.

Comments and Suggestions for Authors

  1. Overall this study has some valuable improvements for the synthesis of the chiral natural product. 
  2. In supporting Information, if the redundant S28-S38 are substituted by tables of selected bond lengths and angles for the crystal structures, their stereochemistry would be easier understood.

Answer: For all crystalline compounds, tables containing selected bond lengths and angles were considered. Figures of crystalline results were also considered.

  1. In the Materials and Methods section, it is easier to follow if putting name and number of the compound to each paragraph, instead of only listing the type of reactions (such as only  "Regiospecific oxirane opening" title).

Answer: In the material and methods section, we put the name and number of the compound in each paragraph, instead of only listing the type of reactions.

Comments on the Quality of English Language

Overall the language is acceptable, with some typos, words and grammars needed to be corrected.

for examples:

Line 95, HBr was added: what form of HBr, gaseous or aqueous? 

Line 83, 0 vs 0 o

line 88, 111,116 etc, remotion vs removal.

Line 96, rotaters the since ?

line 98, compound 3 crystallized vs crystalline compound 3, or crystallized compound 3.

Line 113, 4 was obtained vs 4 was also obtained.

line 149, employ vs use; tested vs explored; line 160, despite the vs although; 

S9 typo, temeratura vs temperature.

Answer: all typos, words, and grammar mistakes were corrected.

Reviewer 3 Report

The manuscript "Chiral indolizinium salts derived from 2-pyridinecarbaldehyde. First diastereoselective syntheses of (–)-1-epi-lentiginosine" reports a new total synthesis strategy of a chiral compound. The design of the synthesis is based on strategies reported earlier by the authors. I think this manuscript is suitable for acceptance after minor revision.

1. Line 15: “stars” should be deleted.

2. Line 17: trans instead of tras.

3. Scheme 2: LG should be disclosed.

4. Line 81: minor instead of minnor.

5. Scheme 2: commercially available reagents should be added.

6. Scheme 3: yields should be added.

7. Lines 87-90: what methods did you test? This should be briefly discussed in the text.

Author Response

Comments and Suggestions for Authors

The manuscript "Chiral indolizinium salts derived from 2-pyridinecarbaldehyde. First diastereoselective syntheses of (–)-1-epi-lentiginosine" reports a new total synthesis strategy of a chiral compound. The design of the synthesis is based on strategies reported earlier by the authors. I think this manuscript is suitable for acceptance after minor revision.

  1. Line 15: “stars” should be deleted.
  2. Line 17: trans instead of tras.
  3. Scheme 2: LG should be disclosed.
  4. Line 81: minor instead of minnor.
  5. Scheme 2: commercially available reagents should be added.
  6. Scheme 3: yields should be added.
  7. Lines 87-90: what methods did you test? This should be briefly discussed in the text.

Answer: all typos, words, and grammar mistakes were corrected.

Reviewer 4 Report

Teran et al reported a novel method foe the synthesis of diastereoselective synthesis of (–)-1-epi-lentiginosine. The key strategy is the chiral auxiliaries assisted diastereoselective synthesis aryl epoxy amides. It’s developed by the authors. Then through 5 steps, the desired product can be obtained in 49.7% overall yield. This paper can be accepted in Molecules if the authors address the following issues:

1.     The product is racemic, so it should be (±)-1-epi-lentiginosine, not (–)-1-epi-lentiginosine

2.     Synthesis of compound 10a and b is a key step, the author should talk about more this step, explain why the particular Yb(OTf)3 was used and give the cited literature.

3.     Since 7a,7b and 10a, 10b can’t be separated in after chromatographic purification and the NMR spectra are given in the supporting information, author should explain the given dr value.

4.     In the supporting information, the yield of 2a+2b is 85%, and the dr is 95:5, the author give the exact yield of 2a in the supporting information.

In summary, this paper can be accepted after minor revision.

This paer is well writen.

Author Response

Comments and Suggestions for Authors

Teran et al reported a novel method foe the synthesis of diastereoselective synthesis of (–)-1-epi-lentiginosine. The key strategy is the chiral auxiliaries assisted diastereoselective synthesis aryl epoxy amides. It’s developed by the authors. Then through 5 steps, the desired product can be obtained in 49.7% overall yield. This paper can be accepted in Molecules if the authors address the following issues:

  1. The product is racemic, so it should be (±)-1-epi-lentiginosine, not (–)-1-epi-lentiginosine

Answer: We report the diastereoselective synthesis of (–)-1-epi-lentiginosine.

  1. Synthesis of compound 10a and b is a key step, the author should talk about more this step, explain why the particular Yb(OTf)3 was used and give the cited literature.

Answer: The following reference was added: Zapata-Machin, E.; Castellan, T.; Baudoin-Dehoux, C.; Génisson, Y. Synth. Commun., 2015, 45, 645-652

  1. Since 7a,7b and 10a, 10b can’t be separated in after chromatographic purification and the NMR spectra are given in the supporting information, author should explain the given dr value.

Answer: The following note was added; (the diastereomeric ratio was measured directly from the NMR spectra of the crude reaction).

  1. In the supporting information, the yield of 2a+2b is 85%, and the dr is 95:5, the author give the exact yield of 2a in the supporting information.

Answer: the correct yield of 2a was added in the supporting information.

In summary, this paper can be accepted after minor revision.